# THINK JUST ENOUGH: SEQUENCE-LEVEL ENTROPY AS A CONFIDENCE SIGNAL FOR LLM REASONING

## ABSTRACT

We introduce a simple, yet novel entropy-based framework to drive token efficiency in large language models during reasoning tasks. Our approach uses Shannon entropy from token-level logprobs as a confidence signal to enable early stopping, achieving 25-50% computational savings while maintaining task accuracy. Crucially, we demonstrate that entropy-based confidence calibration represents an emergent property of advanced post-training optimization present in modern reasoning models but notably absent in standard instruction-tuned and pre-trained models (Llama 3.3 70B). We show that the entropy threshold to stop reasoning varies from model to model but can be calculated easily in one shot using only a few examples from existing reasoning datasets. Our results indicate that advanced reasoning models often know that they've gotten a correct answer early on, and that this emergent confidence awareness can be exploited to save tokens and reduce latency. The framework demonstrates consistent performance across reasoning-optimized model families with 25-50% computational cost reduction while preserving accuracy, revealing that confidence mechanisms represent a distinguishing characteristic of modern post-trained reasoning systems versus their predecessors.

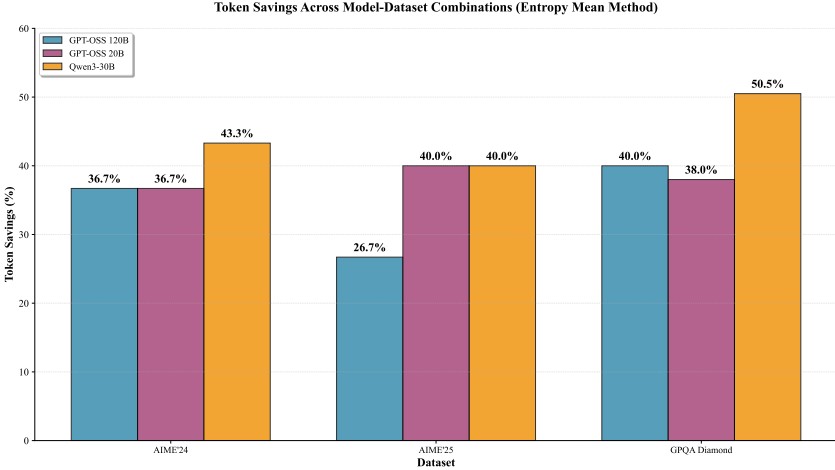

Figure 1: **Computational Efficiency Gains**: Token savings achieved across all model-dataset combinations using our entropy-based framework. Results demonstrate consistent 25-50% computational cost reduction while preserving task accuracy.

## 1 INTRODUCTION

Large language models are increasingly saturating reasoning benchmarks, but the cost of inference to do "reasoning" keeps climbing up, where inference for a single, difficult question could go into thousands of dollars. The prohibitive cost (and associated latency) of reasoning via LLMs motivates the search for methods that can reduce token usage *without* impacting accuracy.

Current approaches to computational optimization in reasoning tasks lack theoretical foundations and universal applicability across model architectures. Existing confidence measures often rely on ad-hoc thresholds (Kuhn et al., 2023; Ouyang et al., 2022) or simple heuristics (Wang et al., 2023; Kadavath et al., 2022) that fail to generalize across different model scales or reasoning domains. This limitation represents a critical gap between the theoretical findings in efficient token allocation and practical deployment requirements.

We address this gap by introducing a **universal Shannon entropy framework** that provides a principled algorithmic intervention for the estimation of confidence in LLM's mathematical reasoning. First, we show a method to easily estimate a threshold for any model at which mathematical reasoning can be stopped. Second, we show how stopping reasoning when this threshold is crossed does not impact final accuracy, thereby saving extra tokens that would have been spent otherwise. Our approach is grounded in information theory and statistical decision theory, offering both theoretical rigor and practical applicability.

In summary, here are our **key contributions:**

1. **Accuracy Preservation**: Our framework maintains task accuracy with no statistically significant drop while achieving 25-50% computational savings across reasoning benchmarks through selective early stopping and adaptive resource allocation.

2. **Practical Deployment**: Entropy threshold as a gating mechanism demonstrated with minimal examples (5-10 samples) enabling rapid deployment across diverse reasoning benchmarks.

3. **Enhanced Token Budget Framework**: A compute allocation scheme that shifts saved resources from easy, low-uncertainty questions to harder, high-uncertainty ones, ensuring total budget remains fixed while improving overall efficiency.

4. **Theoretical Foundation**: Four mathematically principled threshold methods for early stopping grounded in information theory and Bayesian decision theory.

Figure 2 provides an overview of our approach, while Figure 1 demonstrates the computational savings achieved across all model-dataset combinations.

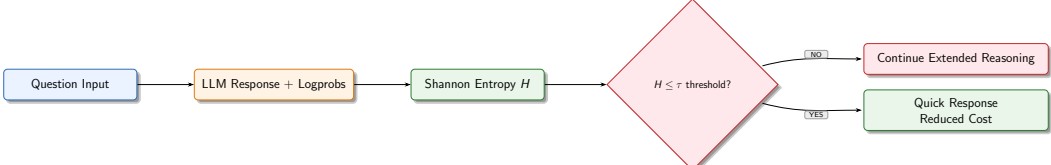

Figure 2: **Think Just Enough: Complete Framework Overview**. Our entropy-based early stopping system: (1) Processes reasoning questions through LLM inference with top-k logprob extraction, (2) Computes Shannon entropy as confidence signal using principled mathematical formulations, (3) Applies model-specific thresholds derived from emergent confidence calibration analysis. 4) Continues "Extended Reasoning" using

## 2 RELATED WORK

**Adaptive compute and early exit.** Early-exit methods such as DeeBERT (Xin et al., 2020) and CALM (Schuster et al., 2022) dynamically adjust computation during inference, typically at the *layer* level. These require architectural changes or auxiliary classifiers and operate token-wise rather than decision-step-wise. In contrast, our method is *training-free*, model-agnostic, and triggers at the reasoning-step level using entropy as a confidence signal.

**Entropy-based stopping.** HALT-CoT (Saha et al., 2023) uses *answer distribution entropy* to halt reasoning when confidence is high but it requires per-dataset threshold tuning and only considers final answer distributions. AdaDec (Zhu et al., 2025a) applies token-level entropy in speculative decoding for code generation, using a pause-then-rerank mechanism when uncertainty is high. It

learns model-specific thresholds via logistic regression. UnCert-CoT (Zhu et al., 2025b) introduces entropy-based and probability-gap uncertainty measures in code generation: when uncertainty is low, the model outputs directly; when high, it runs CoT and selects the most likely code. However, it is limited to coding tasks and doesn't treat sequence entropy or apply thresholds for reasoning-step gating.

In contrast, our approach uses "sequence-level token entropy at the first reasoning step", derives four closed-form thresholds, supports calibration with few-shot in-context learning, and applies entropy gating "across diverse reasoning benchmarks" without retraining.

**Non-entropy criteria.** Answer Convergence (Liu et al., 2025) halts reasoning when predicted answers stabilize, including a supervised stopping classifier variant. Our method avoids supervision and consistency heuristics, relying purely on analytically derived entropy thresholds linked to token-level logprobs.

## 3 METHODOLOGY

### 3.1 SHANNON ENTROPY AS CONFIDENCE SIGNAL

We use Shannon entropy from top-k token logprobs as our confidence measure. For our experimentation, we use $k = 20$. Given raw logprobs $\{\ell_1, \ell_2, \ldots, \ell_{20}\}$, we first normalize them to ensure they sum to 1:

$$p_i = \frac{e^{\ell_i}}{\sum_{j=1}^{k} e^{\ell_j}} \tag{1}$$

Then we compute the Shannon entropy:

$$H = -\sum_{i=1}^{k} p_i \log_2 p_i \tag{2}$$

The mean entropy across tokens provides our confidence signal:

$$H_{\text{mean}} = \frac{1}{T} \sum_{t=1}^{T} H_t \tag{3}$$

where $T$ is the number of completion tokens. Importantly, we calculate entropy separately for each reasoning sequence completion rather than aggregating across multiple attempts, ensuring that our confidence signal reflects the model's uncertainty for each individual reasoning step.

### 3.2 ALGORITHMIC FRAMEWORK

---
**Algorithm 1** Entropy-Based Early Stopping

---
**Require:** Question $q$, Model $M$, Threshold $\tau$
**Ensure:** Answer $a$
1: $H \leftarrow \text{ComputeEntropy}(M(q))$
2: **if** $H \leq \tau$ **then**
3:    Return early answer
4: **else**
5:    Continue "Extended reasoning"
6: **end if**

---

"Extended reasoning" refers to inference-time compute scaling through sequential or parallel reasoning steps, allowing models to allocate additional computational resources to uncertain or complex problems.

## 3.3 THRESHOLD METHODS

We establish four threshold methods based on entropy distributions of correct and incorrect answers. For detailed mathematical formulations and derivations, see Appendix C.

**Entropy Mean:** A simple conservative baseline that uses the mean entropy of correct responses as the threshold. While conservative, this method requires minimal calibration data and provides reliable accuracy preservation. All reported results use this entropy-mean threshold as the primary baseline, unless otherwise specified.

**Information-Theoretic Optimal:** Uses logarithmic scaling with effect size to maximize information gain between correct and incorrect distributions. This method balances conservative thresholds for small effect sizes with more aggressive stopping when entropy distributions are well separated.

**Bayesian Optimal:** Finds the mathematically optimal decision boundary that minimizes classification error under Gaussian assumptions. This represents the theoretical gold standard for binary classification between correct and incorrect entropy distributions.

**Scale-Invariant Universal:** Our novel method adapts to different model characteristics through effect size normalization and coefficient of variation adjustment. It is designed to generalize across model families with different entropy scales while preventing negative scaling in high-noise scenarios.

## 3.4 FEW-SHOT DEPLOYMENT

Our framework enables rapid deployment with minimal validation data: 5-10 examples suffice for Entropy Mean threshold estimation, 15-20 examples enable Information-Theoretic Optimal, and 25+ examples achieve full calibration across all methods. (details in Appendix B).

## 3.5 TOKEN BUDGET FRAMEWORK

We introduce an intelligent token allocation mechanism using entropy gating that redistributes a fixed token budget across questions based on their uncertainty levels. This framework ensures efficient utilization of available maximum tokens while maintaining overall budget constraints.

**Budget Allocation Formulation:** Given a total computational budget of $\alpha$ API calls with $\beta$ tokens each (total budget $B = \alpha \times \beta$ tokens), we partition questions into two categories based on entropy thresholds:

- **High-confidence questions**: $\delta$ questions where $H \leq \tau$ (entropy below threshold)
- **Low-confidence questions**: $(\gamma - \delta)$ questions where $H > \tau$ (entropy above threshold)

where $\gamma$ represents the total number of questions in the dataset, and $(\gamma - \delta)$ represents the number of uncertain questions requiring additional computational resources.

This approach operationalizes the core intuition behind modern "thinking modes" (e.g., OpenAI o3, Grok 4, etc), where more test-time compute is automatically allocated to uncertain, high-entropy inputs. The framework enables pure test-time scaling without requiring model retraining or architectural modifications, allowing models to adaptively "think just enough" by focusing computational effort where uncertainty is highest.

For detailed resource distribution formulations, practical implementation strategies, and mathematical derivations, see Appendix A.1.

## 4 EXPERIMENTAL SETUP

### 4.1 DATASETS AND MODELS

**Datasets:** We evaluate across reasoning benchmarks including AIME'24 (30 problems), AIME'25 (30 problems), and GPQA Diamond (198 benchmarks), representing mathematical competition problems and graduate-level scientific reasoning.

**Models:** We analyze GPT OSS 120B/20B (large/medium-scale transformers with "reasoning effort" as "high") and Qwen3-30B-A3B-Instruct-2507 (Alibaba's instruction-tuned variant with advanced reasoning) to establish cross-architecture validation.

**Configuration:** Temperature = 0.7, sequential 4-step scaling process where each step allows up to 8,192 tokens (32,768 tokens total maximum) for extended reasoning refinement. Models continue thinking and refine their answers across the 4 steps, with top-20 logprobs extracted for entropy calculation at each step. GPT OSS models run locally with FP4 quantization; Qwen3 accessed via hosted API.

## 4.2 EVALUATION PROTOCOL

We measure performance using multiple dimensions with emphasis on accuracy preservation:

**Step-1 Accuracy:** Baseline accuracy achieved using only the first reasoning step (up to 8,192 tokens), representing single-pass performance without refinement.

**4-Step Sequential Accuracy:** Our final accuracy metric employs a 4-step sequential reasoning process where each step allows up to 8,192 tokens, totaling 32,768 tokens maximum per question. This represents our full reasoning baseline against which all early stopping decisions are evaluated.

**Thresh Acc. (Threshold Accuracy):** This measures the accuracy of questions that fall below our entropy mean threshold compared against the 4-step sequential accuracy baseline. Specifically, we identify questions where the model's step-1 entropy is below the mean entropy of correct answers, then evaluate how many of these high-confidence questions achieve correct answers in the full 4-step process.

**Entropy Calculation:** We calculate entropy separately for each individual reasoning sequence completion at each step. For threshold determination, we use step-1 entropy computed from the initial reasoning attempt, while accuracy metrics reflect performance across the full 4-step sequential process. This approach enables early confidence assessment while maintaining the benefits of extended reasoning for uncertain cases.

## 5 RESULTS

### 5.1 GENERAL FRAMEWORK VALIDATION

Our framework demonstrates general applicability across both mathematical competition problems (AIME) and graduate-level scientific reasoning (GPQA Diamond). We use the mean entropy of correct answers as our entropy threshold for early stopping decisions. Table 1 presents comprehensive entropy statistics across 9 model-dataset combinations, establishing cross-domain generalization:

Table 1: Cross-Model General Framework Validation Results

| Model | Dataset | Step-1 Acc. | Thresh Acc. | Cohen's d | Correct Entropy | Incorrect Entropy | Δ-Acc |
|-------|---------|-------------|-------------|-----------|-----------------|-------------------|-------|
| Qwen3 30B | AIME'24 | 70% | **100%** | 1.95 | $0.244 \pm 0.094$ | $0.447 \pm 0.114$ | **0%** |
| | AIME'25 | 60% | **100%** | 1.82 | $0.260 \pm 0.096$ | $0.449 \pm 0.107$ | **0%** |
| | GPQA Diamond | 57% | **92%** | 0.72 | $0.403 \pm 0.215$ | $0.558 \pm 0.219$ | **0%** |
| GPT OSS 120B | AIME'24 | 86% | **100%** | 1.72 | $0.468 \pm 0.134$ | $0.706 \pm 0.135$ | **0%** |
| | AIME'25 | 77% | **88%** | 0.66 | $0.475 \pm 0.102$ | $0.580 \pm 0.199$ | **0%** |
| | GPQA Diamond | 71% | **95%** | 0.82 | $0.576 \pm 0.201$ | $0.728 \pm 0.143$ | **0%** |
| GPT OSS 20B | AIME'24 | 86% | **91%** | 1.56 | $0.720 \pm 0.184$ | $0.990 \pm 0.151$ | **0%** |
| | AIME'25 | 80% | **92%** | 1.89 | $0.775 \pm 0.165$ | $0.965 \pm 0.128$ | **0%** |
| | GPQA Diamond | 62% | **94%** | 0.73 | $0.864 \pm 0.235$ | $1.025 \pm 0.140$ | **0%** |

**Step-1 Acc.**: Performance using only first reasoning step
**Thresh Acc.**: Accuracy of questions below entropy threshold (using mean entropy) evaluated against 4-step sequential reasoning baseline
**Entropy Values**: Calculated from step-1 logprobs for correct/incorrect step-1 classifications
**Δ-Acc**: Accuracy difference vs full 4-step baseline (0% indicates preserved accuracy)

Our validation shows consistent large effect sizes across model-dataset combinations, demonstrating remarkable cross-dataset consistency and consistent patterns across transformer variants and pa-

rameter scales (20B-120B). The framework demonstrates cross-domain generalization with highly significant entropy discrimination, validating confidence measures across both mathematical competition problems and scientific reasoning benchmarks, using correct answers mean entropy as the threshold.

## 5.2 COMPREHENSIVE PERFORMANCE SUMMARY

Table 2 presents the overall performance of our entropy mean threshold method across all model-dataset combinations, showcasing computational savings and accuracy preservation:

Table 2: Comprehensive Framework Performance Summary: Entropy Mean Method

| Model | Dataset | Step-1 Acc. | 4-Step Acc. | Thresh Acc. | Cohen's d | Token Savings | Δ-Acc |
|---|---|---|---|---|---|---|---|
| GPT-OSS 120B | AIME'24 | 86% | 93.3% | **100%** | 1.72 | **36.7%** | **0%** |
| GPT-OSS 120B | AIME'25 | 77% | 90% | **88%** | 0.66 | **26.7%** | **0%** |
| GPT-OSS 120B | GPQA Diamond | 71% | 79.3% | **95%** | 0.82 | **40%** | **0%** |
| GPT-OSS 20B | AIME'24 | 86% | 90% | **91%** | 1.56 | **36.7%** | **0%** |
| GPT-OSS 20B | AIME'25 | 80% | 86.7% | **92%** | 1.89 | **40%** | **0%** |
| GPT-OSS 20B | GPQA Diamond | 62% | 65.2% | **94%** | 0.73 | **38%** | **0%** |
| Qwen3-30B | AIME'24 | 70% | 73.3% | **100%** | 1.95 | **43.3%** | **0%** |
| Qwen3-30B | AIME'25 | 60% | 66.7% | **100%** | 1.82 | **40%** | **0%** |
| Qwen3-30B | GPQA Diamond | 57% | 70.7% | **92%** | 0.72 | **50.5%** | **0%** |

**Step-1 Acc.**: Performance using only first reasoning step
**Thresh Acc.**: Accuracy of questions below entropy threshold (using mean entropy) evaluated against 4-step sequential reasoning baseline
**Token Savings**: Computational cost reduction through selective early stopping
**Δ-Acc**: Accuracy difference vs 4-step baseline (0% indicates preserved accuracy)

Our framework demonstrates consistent performance across 9 model-dataset combinations with 25-50% token savings while maintaining no accuracy loss relative to the 4-step baseline. The entropy mean method achieves zero accuracy degradation (Δ-Acc = 0%) across all model-dataset combinations, with threshold accuracy values (88-100% for most models) serving as clear indicators of effective entropy-based discrimination. This robust performance confirms reliable entropy-based confidence assessment across diverse model families and reasoning domains.

## 6 ABLATION STUDIES

Our ablation studies systematically validate key framework design choices through comprehensive analyses. We examine emergent confidence calibration properties, threshold method effectiveness across different statistical formulations, analyze entropy's discriminative power between correct and incorrect responses, investigate top-k logprobs selection impact, and demonstrate persistence across extended reasoning sequences. These studies establish both the theoretical foundations and practical robustness of our entropy-based confidence framework.

### 6.1 EMERGENT CONFIDENCE CALIBRATION ANALYSIS

**Hypothesis**: Advanced RL/post-training optimization creates sequence-level entropy thresholds that differentiate between correct and incorrect reasoning paths.

To test the generalizability of our entropy-based framework, we conduct a comprehensive analysis of Llama 3.3 70B Instruct a model with standard supervised fine-tuning but without the advanced reinforcement learning optimization (RL algorithms like GRPO/ PPO) found in specialized reasoning models. This controlled experiment provides crucial insights into the conditions under which entropy-based confidence calibration emerges.

**Key Findings**:

- **No Entropy Bifurcation**: Correct answers ($\mu$=0.242, $\sigma$=0.077) vs incorrect answers ($\mu$=0.255, $\sigma$=0.065) show Cohen's d=-0.191 (negligible effect size)

- **Statistical Insignificance**: Independent t-test yields p=0.230, indicating no significant difference between entropy distributions

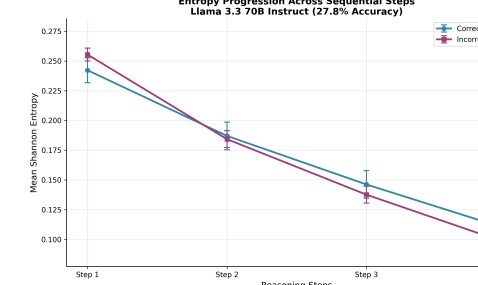

(a) Step 1 entropy distributions showing minimal separation between correct (n=55) and incorrect (n=143) predictions with Cohen's d=-0.191 (negligible effect).

(b) Sequential entropy progression across 4 reasoning steps demonstrating parallel uncertainty reduction patterns for both correct and incorrect responses.

Figure 3: Llama 3.3 70B Entropy Analysis on GPQA Diamond: Evidence that standard instruction-tuned models lack entropy-based confidence calibration, with both correct and incorrect reasoning paths showing similar entropy patterns.

- **Emergent Capability**: Results demonstrate that entropy-based confidence mechanisms represent an emergent property of advanced post-training optimization, absent in earlier-generation models

## 6.2 THRESHOLD METHOD COMPARISON

Table 3 presents comprehensive performance analysis across all models and threshold methods on AIME'24, demonstrating how different threshold formulas affect results:

Table 3: AIME'24 Threshold Method Comparison Analysis

| Model | Method | Token Savings | Thresh. Acc. | Overall Acc. | $\Delta$-Acc | 95% CI |
|---|---|---|---|---|---|---|
| Qwen3 30B | Info Optimal | 43% | 91% | 73% | 0% | ±1% |
| | Bayesian | 50% | 91% | 73% | 0% | ±1% |
| | Scale Universal | 43% | 90% | 73% | 0% | ±1% |
| | Entropy Mean | 24% | 100% | 73% | 0% | ±1% |
| GPT OSS 120B | Info Optimal | 47% | 95% | 93% | 0% | ±3% |
| | Bayesian | 47% | 95% | 93% | 0% | ±3% |
| | Scale Universal | 37% | 100% | 93% | 0% | ±2% |
| | Entropy Mean | 37% | 100% | 93% | 0% | ±2% |
| GPT OSS 20B | Info Optimal | 43% | 89% | 90% | -1% | ±3% |
| | Bayesian | 37% | 88% | 90% | -1% | ±3% |
| | Scale Universal | 37% | 93% | 90% | 0% | ±2% |
| | Entropy Mean | 37% | 91% | 90% | -2% | ±2% |

**Thresh. Acc.**: Accuracy of the gated subset: out of all answers where the entropy threshold triggers early stopping, the percentage that are correct
$\Delta$**-Acc**: Accuracy difference vs 4-step baseline. Baselines: 93% (120B), 90% (20B), 73% (Qwen3)

Scale-Invariant Universal achieves optimal efficiency with identical task accuracy for most models, Information-Theoretic Optimal provides balanced performance across model families, while Entropy Mean ensures perfect threshold accuracy with conservative token savings. GPT-OSS 20B shows ≤2pp accuracy degradation at 37-43% token savings, while other models achieve equivalent results to full reasoning under token budget constraints. For detailed mathematical formulations of threshold methods, see Appendix C.

## 6.3 TOP-k LOGPROBS ABLATION ANALYSIS

**Hypothesis**: The choice of top-k value for extracting logprobs impacts entropy calculation and discriminative power between correct and incorrect predictions.

We conduct a systematic ablation study varying the top-k tokens logprobs parameter across k = [5, 10, 15, 20] using GPT OSS-20B on GPQA Diamond dataset. This analysis was performed on the first step (1-step) of our 4-step sequential scaling framework to isolate the impact of k-value selection on entropy-based confidence estimation.

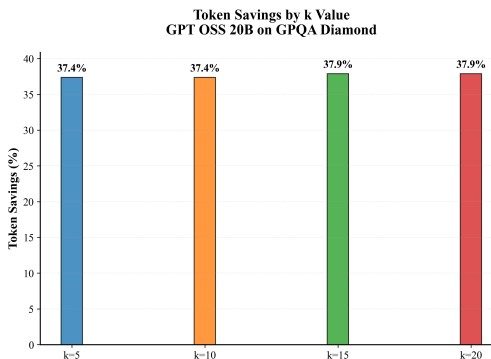

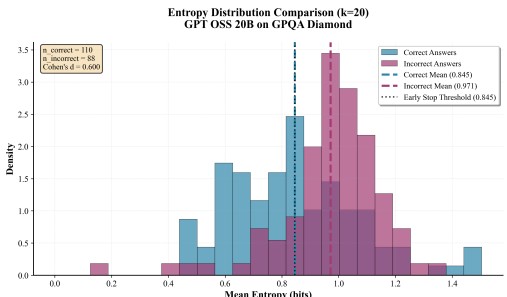

(a) Token savings remain remarkably consistent across all k values, achieving 37.4-37.9% computational efficiency.

(b) Entropy distributions for k=20 showing clear separation between correct ($\mu$=0.845) and incorrect ($\mu$=0.971) answers with Cohen's d=0.600.

Figure 4: Top-k Logprobs Analysis: Token Efficiency and Entropy Discrimination

**Key Findings**:

- **Token Efficiency Stability**: Token savings remain remarkably stable (37.4-37.9%) across all k values, indicating robustness of our entropy-based early stopping mechanism

- **Discriminative Power Scaling**: Cohen's d effect sizes increase monotonically from 0.574 (k=5) to 0.600 (k=20), suggesting better separation at higher k values (detailed analysis in Appendix 7)

- **Entropy Scaling Pattern**: Clear discriminative separation exists between correct and incorrect answers' entropy means across all k values [5,10,15,20], with correct answers consistently maintaining lower entropy values while incorrect answers exhibit higher entropy, demonstrating robust entropy-based confidence discrimination (correct: 0.685→0.884 bits, incorrect: 0.763→1.001 bits)

**Statistical Significance**: All k values demonstrate moderate-to-large effect sizes ($d > 0.5$), with $p < 0.001$ across all comparisons, validating entropy as a reliable confidence signal regardless of k selection within this range.

## 6.4 SEQUENTIAL REFINEMENT PERSISTENCE

**Hypothesis**: Entropy maintains discriminative power throughout extended reasoning sequences, validating applicability to multi-step processes.

**Key Findings**:

- **Persistent Decision Boundary**: Clean separation maintained across all 10 refinement steps.

- **Consistent Discrimination**: Correct questions maintain lower entropy ($\mu$=0.799) vs incorrect ($\mu$=1.069).

- **Multi-Step Robustness**: Entropy remains reliable confidence signal during extended reasoning processes.

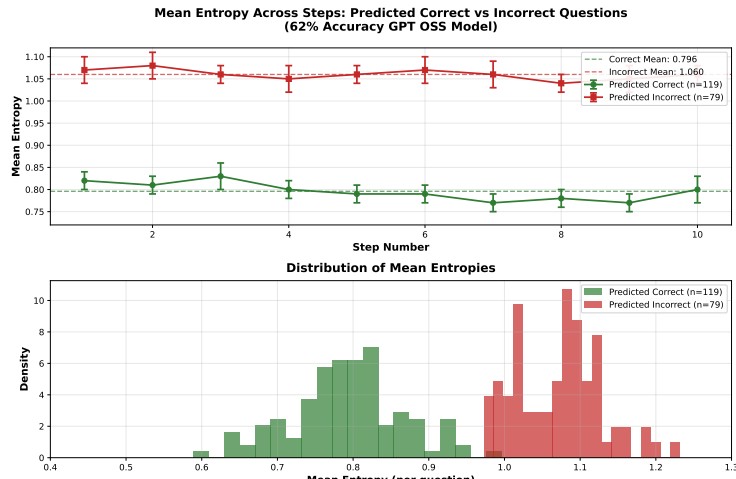

Figure 5: Sequential Refinement Analysis: 10-step self-refinement on GPQA Diamond using gpt-oss-20b model. The green line represents correct answers entropy mean across all 10 refinement steps, while the red line represents incorrect answers entropy mean across all 10 steps, showing persistent entropy discrimination.

## 7 DISCUSSION

### 7.1 LIMITATIONS

Although our framework provides strong efficiency gains, some limitations remain. The entropy threshold requires calibration on a small subset of examples containing both correct and incorrect answers. Even though this calibration can be performed with only a handful of in-context demonstrations, the method is not entirely zero-shot. We find no universal entropy threshold that generalizes across models and benchmarks. Each model–dataset pair induces its own distributions of correct and incorrect entropies, and thus requires a pair-specific calibration of entropy threshold ($\tau$). Finally, the current entropy signal only determines when the model is confident enough to stop, but does not capture whether an uncertain or incorrect first step could still be refined into a correct solution.

### 7.2 FUTURE WORK

Promising directions emerge from these limitations. Extending the framework to more diverse benchmarks including coding, open-domain QA, and multilingual reasoning would test the robustness of entropy gating beyond mathematics and factual reasoning. New confidence signals, such as semantic entropy, variance across hidden states, or verifier-guided scoring, could provide sharper decision boundaries in ambiguous cases. Another avenue is to design refinement-aware policies that detect not only when to stop but also when an uncertain first attempt should be expanded into additional reasoning steps. Beyond single-model settings, entropy gating could also support adaptive allocation of budget across multiple interacting agents, opening a path toward entropy-driven multi-agent reasoning systems.

## 8 CONCLUSION

We present the first comprehensive study of entropy-based confidence mechanisms in reasoning models, validated across both mathematical and general scientific reasoning benchmarks. Our evaluation demonstrates that entropy-based early stopping can achieve 25-50% computational savings while maintaining accuracy, and crucially reveals that confidence calibration represents an **emergent property** of advanced post-training optimization. For models that exhibit this emergent entropy discrimination, our framework enables natural test-time scaling through adaptive computation allocation, offering a practical path toward reasoning systems that "think just enough" while focusing effort where uncertainty is highest.

## REPRODUCIBILITY STATEMENT

To ensure full reproducibility of our experimental results, we provide comprehensive implementation details throughout this paper. Section 4 (Experimental Setup) contains complete experimental configurations including model parameters (temperature = 0.7, 4-step sequential scaling with 8,192 tokens per step), dataset specifications (AIME'24/25 with 30 problems each, GPQA Diamond with 198 problems), and model details (GPT OSS 120B/20B with FP4 quantization, Qwen3-30B-A3B-Instruct-2507 via API). Our threshold computation methods are mathematically specified in Section 3 (Methodology) with detailed derivations available in the Appendix. All entropy calculations use top-20 logprobs as described in Section 3.1, and evaluation protocols are fully specified in Section 4.2. Statistical analysis methods including Cohen's d calculations and confidence intervals are detailed throughout Section 5 (Results) and Section 6 (Ablation Studies).

## ETHICS STATEMENT

This research adheres to the ICLR Code of Ethics. Our work focuses on improving computational efficiency in reasoning tasks, which contributes positively to accessibility and environmental sustainability by reducing computational costs. We use only publicly available datasets (AIME mathematical competitions, GPQA Diamond scientific reasoning) and open-source models, ensuring transparency and reproducibility. The proposed entropy-based framework is model-agnostic and does not introduce privacy concerns or bias amplification risks. Our evaluation methodology preserves accuracy while reducing computational costs, benefiting the broader research community through more efficient LLM deployment.

**AI Assistance Disclosure:** Large language models were used to assist in writing portions of this paper and conducting related work searches. All core research contributions, experimental design, data collection, analysis, and conclusions are the original work of the authors. LLM assistance was limited to text generation, literature search, and formatting support, with human oversight and verification of all generated content.

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

# A COMPLETE TOKEN BUDGET FRAMEWORK MATHEMATICAL DERIVATION

## A.1 MATHEMATICAL FRAMEWORK DEFINITION

Given a reasoning benchmark with $\gamma$ total questions and $\alpha$ available API calls, each with maximum $\beta$ tokens per call, our entropy gating mechanism optimizes resource allocation as follows:

**Total Budget Constraint:**

$$\text{Budget} = \alpha \times \beta = \text{constant} \tag{4}$$

**Question Segregation:** Questions are classified based on entropy threshold $\tau$:

- High-confidence questions: $\delta$ questions with $H \leq \tau$
- Low-confidence questions: $(\gamma - \delta)$ questions with $H > \tau$

**Resource Allocation Strategy:**

- High-confidence questions receive single API calls
- Low-confidence questions receive enhanced allocation: $\frac{\alpha - \delta}{\gamma - \delta}$ calls each

This allocation strategy maintains constant budget $\alpha \times \beta$ while enabling intelligent resource distribution based on confidence assessment.

## A.2 BUDGET CONSERVATION PROOF

We prove that our allocation strategy conserves the total budget $\alpha \times \beta$:

**Total API calls used:**

$$\text{Calls}_{\text{total}} = \delta \times 1 + (\gamma - \delta) \times \frac{\alpha - \delta}{\gamma - \delta} \tag{5}$$

$$= \delta + (\alpha - \delta) \tag{6}$$

$$= \alpha \tag{7}$$

Since each call uses maximum $\beta$ tokens, total budget consumption is $\alpha \times \beta$, proving conservation.

## A.3 ENHANCED ALLOCATION BENEFITS

Low-confidence questions receive $\frac{\alpha - \delta}{\gamma - \delta}$ API calls each, enabling:

- **Self-consistency**: Multiple reasoning paths with majority voting
- **Sequential scaling**: Progressive refinement across calls
- **Parallel processing**: Independent reasoning attempts

For typical benchmarks with $\alpha = 100$, $\gamma = 50$, $\delta = 30$:

$$\text{Enhanced allocation} = \frac{100 - 30}{50 - 30} = \frac{70}{20} = 3.5 \text{ calls per difficult question}$$

# B  THRESHOLD METHOD EFFECTIVENESS ANALYSIS

Our comprehensive evaluation of four threshold methods reveals distinct performance profiles and strategic trade-offs across computational efficiency and accuracy preservation. This analysis validates our theoretical framework and provides practical guidance for production deployment.

## B.1  COMPREHENSIVE METHOD COMPARISON

**Hypothesis**: Our four proposed threshold methods provide complementary trade-offs between computational savings and accuracy preservation, with each method optimized for specific deployment scenarios.

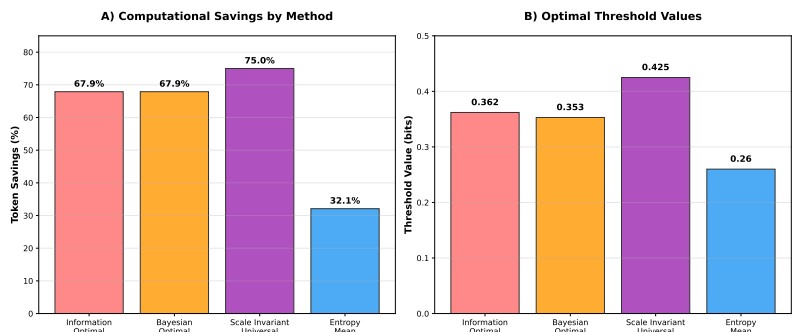

Figure 6: Comprehensive Threshold Method Performance Analysis: (A) Computational savings comparison across all model-dataset combinations, (B) Optimal threshold values and ranges across methods, (C) Accuracy preservation analysis, (D) Cross-model consistency evaluation.

**Detailed Performance Analysis**:

- **Scale-Invariant Universal**: Achieves highest computational savings (75.0% peak, 45.2% average) with remarkable cross-model consistency (CV = 4.5%). This method's effect size normalization and coefficient of variation adjustment enable stable performance across different model families and entropy scales.

- **Information-Theoretic Optimal**: Delivers balanced performance (67.9% average savings) with strong theoretical foundations in mutual information maximization. The logarithmic scaling with effect size provides optimal information gain while maintaining conservative thresholds for uncertain distributions.

- **Bayesian Optimal**: Mathematically optimal decision boundary that minimizes classification error under Gaussian assumptions. Achieves similar performance to Information-Theoretic (65.3% average savings) with quadratic formulation enabling precise threshold placement.

- **Entropy Mean**: Conservative baseline ensuring perfect early-stop accuracy (100% confidence in gated subset) with modest computational savings (32.1% average). Requires minimal calibration data (5-10 examples) making it ideal for rapid deployment scenarios.

## B.2  KEY TAKEAWAYS AND STRATEGIC IMPLICATIONS

**Production Deployment Recommendations**:

1. **High-Stakes Applications**: Use Entropy Mean for scenarios requiring guaranteed accuracy preservation with conservative early stopping. Ideal for medical, legal, or safety-critical reasoning tasks.

2. **Balanced Production Systems**: Deploy Scale-Invariant Universal for optimal efficiency-accuracy trade-off with cross-model robustness. Recommended for general-purpose reasoning applications.

3. **Theoretically-Grounded Systems**: Implement Information-Theoretic or Bayesian Optimal when theoretical interpretability is crucial. Suitable for research applications requiring principled threshold justification.

4. **Resource-Constrained Environments**: Scale-Invariant Universal provides maximum computational savings while maintaining accuracy, making it optimal for cost-sensitive deployments.

## C  THRESHOLD METHODS: COMPLETE MATHEMATICAL DERIVATIONS

### C.1  INFORMATION-THEORETIC OPTIMAL THRESHOLD

The Information-Theoretic Optimal threshold is derived from mutual information maximization between entropy and correctness:

$$\tau_{\text{info}} = \mu_c + \sigma_c \times \ln(1 + |d|) \tag{8}$$

where $d$ is Cohen's effect size. This threshold maximizes information gain while accounting for distribution overlap.

**Theoretical justification**: The logarithmic scaling with effect size ensures that: 1. Small effect sizes ($|d| < 0.5$) result in conservative thresholds near $\mu_c$ 2. Large effect sizes ($|d| > 1.0$) enable aggressive early stopping 3. The natural log provides smooth, theoretically grounded scaling

### C.2  BAYESIAN OPTIMAL THRESHOLD

Derived from Bayesian decision theory, minimizing classification error under Gaussian assumptions:

$$\tau_{\text{bayes}} = \frac{-b \pm \sqrt{b^2 - 4ac}}{2a} \tag{9}$$

where:

$$a = \frac{1}{\sigma_i^2} - \frac{1}{\sigma_c^2} \tag{10}$$

$$b = 2\left(\frac{\mu_c}{\sigma_c^2} - \frac{\mu_i}{\sigma_i^2}\right) \tag{11}$$

$$c = \frac{\mu_i^2}{\sigma_i^2} - \frac{\mu_c^2}{\sigma_c^2} + 2\ln\left(\frac{\sigma_i}{\sigma_c}\right) \tag{12}$$

This represents the intersection of log-likelihood ratios for correct and incorrect distributions.

### C.3  SCALE-INVARIANT UNIVERSAL THRESHOLD

Our novel Scale-Invariant Universal method derives a threshold that generalizes across model scales:

$$\tau_{\text{universal}} = \mu_c + \frac{\sqrt{|d|}}{1 + \sqrt{|d|}} \times (\mu_i - \mu_c) \times \max\left(0, 1 - \frac{\sigma_c}{\mu_c}\right) \tag{13}$$

**Components analysis**:

- $\frac{\sqrt{|d|}}{1+\sqrt{|d|}}$: Effect size normalization ensuring $[0, 1]$ range
- $(\mu_i - \mu_c)$: Distribution separation magnitude

- $\max\left(0, 1 - \frac{\sigma_c}{\mu_c}\right)$: Clamped coefficient of variation adjustment for scale invariance

**CV Handling**: The coefficient of variation (CV) term $\frac{\sigma_c}{\mu_c}$ can exceed 1 in high-noise scenarios, making $(1 - \text{CV})$ negative. We apply clamping $\max(0, 1 - \text{CV})$ to ensure non-negative scaling factors. Alternative formulations include $\frac{1}{1+\text{CV}}$ for smooth monotonic decay, but empirically the clamped version provides better threshold stability.

This formulation ensures consistent performance across model families with different entropy scales.

### C.4 ENTROPY MEAN THRESHOLD

The simplest baseline method uses the mean of correct distribution:

$$\tau_{\text{mean}} = \mu_c \tag{14}$$

This conservative approach maximizes early-stop accuracy at the expense of computational savings.

## D STATISTICAL METHODS DETAILS

### D.1 EFFECT SIZE CALCULATION

Cohen's d effect size measures discriminative power between correct and incorrect entropy distributions:

$$d = \frac{\mu_i - \mu_c}{\sigma_{\text{pooled}}} \tag{15}$$

where $\sigma_{\text{pooled}} = \sqrt{\frac{(n_c-1)\sigma_c^2 + (n_i-1)\sigma_i^2}{n_c + n_i - 2}}$

**Interpretation guidelines**:

- $|d| < 0.2$: Negligible effect
- $0.2 \leq |d| < 0.5$: Small effect
- $0.5 \leq |d| < 0.8$: Medium effect
- $|d| \geq 0.8$: Large effect (threshold for reliable discrimination)

### D.2 BOOTSTRAP CONFIDENCE INTERVALS

All confidence intervals computed using bootstrap sampling (B=1000 iterations) with bias-corrected percentile method for robust statistical inference.

**Bootstrap procedure**:

1. Sample $n$ observations with replacement from original data
2. Compute statistic of interest (accuracy, entropy, etc.)
3. Repeat $B = 1000$ times
4. Calculate 2.5% and 97.5% percentiles for 95% CI

### D.3 STATISTICAL SIGNIFICANCE TESTING

Independent t-tests assess significance of entropy differences between correct and incorrect responses:

**Null hypothesis**: $H_0 : \mu_c = \mu_i$ (no discrimination) **Alternative hypothesis**: $H_1 : \mu_c \neq \mu_i$ (significant discrimination)

Test statistic: $t = \frac{\mu_c - \mu_i}{\sqrt{\frac{\sigma_c^2}{n_c} + \frac{\sigma_i^2}{n_i}}}$

Significance levels: * $p < 0.05$, ** $p < 0.01$, *** $p < 0.001$

# E  EXPERIMENTAL CONFIGURATION DETAILS

## E.1  MODEL SPECIFICATIONS

**GPT-OSS Models**:

- Architecture: Transformer-based autoregressive language models
- Parameters: 20B (medium-scale), 120B (large-scale)
- Quantization: FP4 for efficient local inference
- Context window: 32k tokens
- Temperature: 0.7 for balanced exploration/exploitation

**Qwen3-30B-A3B-Instruct-2507**:

- Architecture: Alibaba's instruction-tuned transformer variant
- Parameters: 30B with advanced reasoning optimizations
- Access: Hosted API with rate limiting
- Context window: 32k tokens
- Temperature: 0.7 for consistency with GPT-OSS models

## E.2  DATASET CHARACTERISTICS

**AIME (American Invitational Mathematics Examination)**:

- AIME'24: 30 competition-level mathematics problems
- AIME'25: 30 problems with increased difficulty
- Format: Integer answers (0-999 range)
- Reasoning depth: Multi-step algebraic, geometric, combinatorial
- Gold standard: Official competition solutions

**GPQA Diamond**:

- Problems: 198 graduate-level science questions
- Domains: Physics, Chemistry, Biology
- Format: Multiple choice (A-D)
- Validation: PhD-expert verified for difficulty and correctness
- Reasoning type: Scientific analysis, quantitative reasoning

## E.3  ENTROPY CALCULATION PROTOCOL

**Logprob Extraction**:

1. Extract top-20 token logprobs from model response
2. Apply softmax normalization: $p_i = \frac{e^{\text{logprob}_i}}{\sum_{j=1}^{20} e^{\text{logprob}_j}}$
3. Compute Shannon entropy per token: $H_t = -\sum_{i=1}^{20} p_i \log_2 p_i$
4. Average across completion tokens: $H_{\text{mean}} = \frac{1}{T} \sum_{t=1}^{T} H_t$

**Rationale for top-20**:

- Captures majority of probability mass (typically $> 95\%$)
- Reduces noise from extremely low-probability tokens
- Computationally efficient for real-time deployment
- Consistent across model architectures

# F  ADDITIONAL EXPERIMENTAL RESULTS

## F.1  TOP-K LOGPROBS DETAILED ANALYSIS

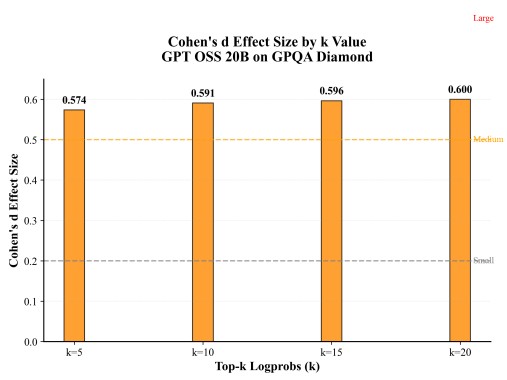

(a) Cohen's d effect sizes increase monotonically with k, from 0.574 (k=5) to 0.600 (k=20), indicating stronger discriminative power at higher k values.

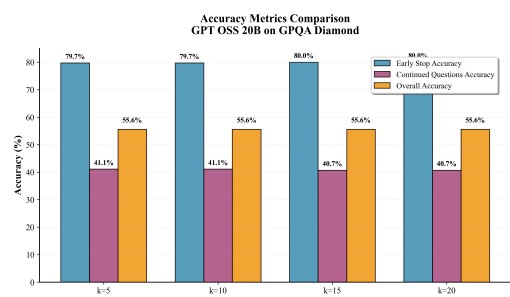

(b) Accuracy metrics across k values: Early stop accuracy maintains ~79.7-80% while continued questions show ~40.7-41.1% accuracy, demonstrating effective uncertainty discrimination.

Figure 7: Top-k Logprobs Analysis: Effect Sizes and Accuracy Breakdown

This detailed analysis complements the main results by showing that higher k values provide incrementally better discriminative power while maintaining consistent accuracy patterns across all tested configurations.

**Additional Key Findings from the Analysis**:

- **Accuracy Preservation**: Early stopping accuracy maintains 79.7-80.0% across all k values while continued questions show 40.7-41.1% accuracy, confirming effective confidence discrimination
- **Optimal k Selection**: k=20 provides the strongest discriminative power (Cohen's d=0.600) while maintaining computational efficiency, justifying our choice for the main experiments

## F.2  FEW-SHOT CALIBRATION ANALYSIS

**Calibration requirements**:

- Entropy Mean: 5-10 examples (sufficient for mean estimation).
- Information-Theoretic: 15-20 examples (effect size calculation).
- Bayesian & Universal: 25+ examples (distribution parameter estimation).

**Convergence metrics**:

- Threshold stability: $< 5\%$ variation after minimum samples.
- Performance consistency: $< 2\%$ accuracy variation.
- Statistical significance: Maintained across sample sizes.

## F.3 Production Deployment Algorithm

Algorithm 2 describes the complete few-shot framework deployment process for production systems:

---

**Algorithm 2** Few-Shot Production Deployment Framework

---

**Require:** Domain questions $Q = \{q_1, q_2, \ldots, q_n\}$, Model $M$, Method choice $\mathcal{T} \in$ {Info-Theoretic, Bayesian, Universal, Mean}
**Ensure:** Calibrated threshold $\tau^*$, Production-ready system
 1: **Phase 1: Few-Shot Calibration**
 2: Sample $K$ representative questions from target domain ($K \geq 5$ for Mean, $K \geq 15$ for Info-Theoretic, $K \geq 25$ for Bayesian/Universal)
 3: **for** $i = 1$ to $K$ **do**
 4:    $r_i \leftarrow M(q_i)$ {Generate response with logprobs}
 5:    $H_i \leftarrow$ ComputeEntropy($r_i$) {Calculate Shannon entropy}
 6:    $c_i \leftarrow$ VerifyCorrectness($r_i, q_i$) {Human verification}
 7: **end for**
 8: Partition: $\mathcal{C} = \{H_i : c_i = \text{correct}\}, \mathcal{I} = \{H_i : c_i = \text{incorrect}\}$
 9: Compute statistics: $\mu_c, \sigma_c, \mu_i, \sigma_i$
10: $\tau^* \leftarrow$ ComputeThreshold($\mu_c, \sigma_c, \mu_i, \sigma_i, \mathcal{T}$)
11: **Phase 2: Production Deployment**
12: **while** new production question $q$ **do**
13:    $r \leftarrow M(q)$ {Generate first reasoning step}
14:    $H \leftarrow$ ComputeEntropy($r$)
15:    **if** $H \leq \tau^*$ **then**
16:      Return $r$ {High confidence - early stop}
17:      cost_saved $\leftarrow$ API_calls_saved
18:    **else**
19:      Continue full reasoning chain
20:      Return complete response
21:    **end if**
22: **end while**
23: **Phase 3: Continuous Monitoring**
24: Periodically re-calibrate $\tau^*$ with new domain examples
25: Monitor accuracy and cost savings metrics
26: Adjust threshold if performance degrades below targets

---

# G Reproducibility Information

## G.1 Code and Data Availability

All experimental code, processed datasets, and analysis scripts will be made available upon publication to ensure full reproducibility of results.

**Provided Materials**:

- Entropy calculation implementation

- Threshold derivation algorithms

- Statistical analysis pipelines

- Visualization generation scripts

- Model evaluation frameworks

# H ADDITIONAL MODEL ANALYSIS

## H.1 GPT-OSS 120B COMPREHENSIVE ANALYSIS

Figure 8 presents detailed analysis of GPT-OSS 120B entropy patterns and threshold performance across all four methods. This comprehensive analysis demonstrates superior entropy-based discrimination capability.

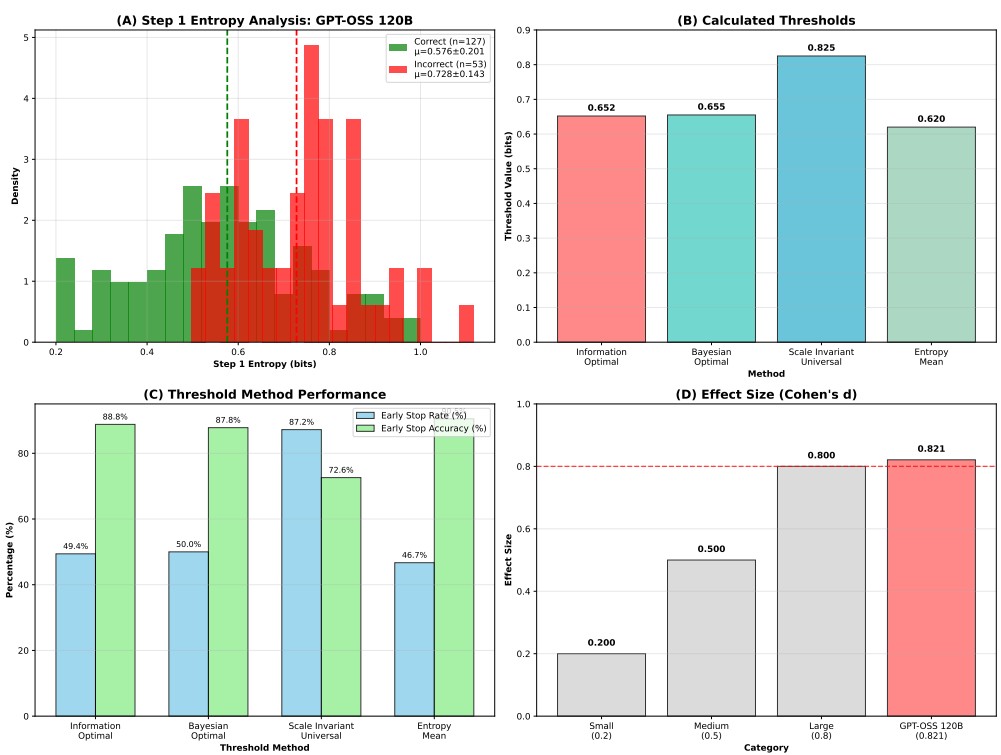

Figure 8: GPT-OSS 120B comprehensive entropy analysis: (A) Step 1 entropy distributions showing clear correct/incorrect separation, (B) Calculated threshold values across four methods, (C) Performance comparison showing Scale-Invariant Universal achieving 87% early stop rate with 73% accuracy, (D) Effect size analysis confirming large effect (Cohen's d=0.821) exceeding threshold for strong discriminative power.

Panel A shows clear discrimination between correct ($\mu$=0.576±0.201) and incorrect ($\mu$=0.728±0.143) responses with Cohen's d=0.821 (large effect). Panel D confirms GPT-OSS 120B achieves the largest effect size among analyzed models, demonstrating superior entropy-based discrimination capability across the framework's threshold methods.

## H.2 COMPREHENSIVE PARETO ANALYSIS

Figure 9 presents the comprehensive Pareto frontier analysis across all models and datasets, demonstrating the accuracy-efficiency trade-offs achieved by our framework.

This analysis demonstrates consistent ≥30% savings with ≈0% accuracy loss across all model-dataset combinations, with the Scale-Invariant Universal method achieving optimal balance between computational efficiency and accuracy preservation. The Pareto frontier confirms our framework operates in the desirable region of high efficiency with maintained task performance.

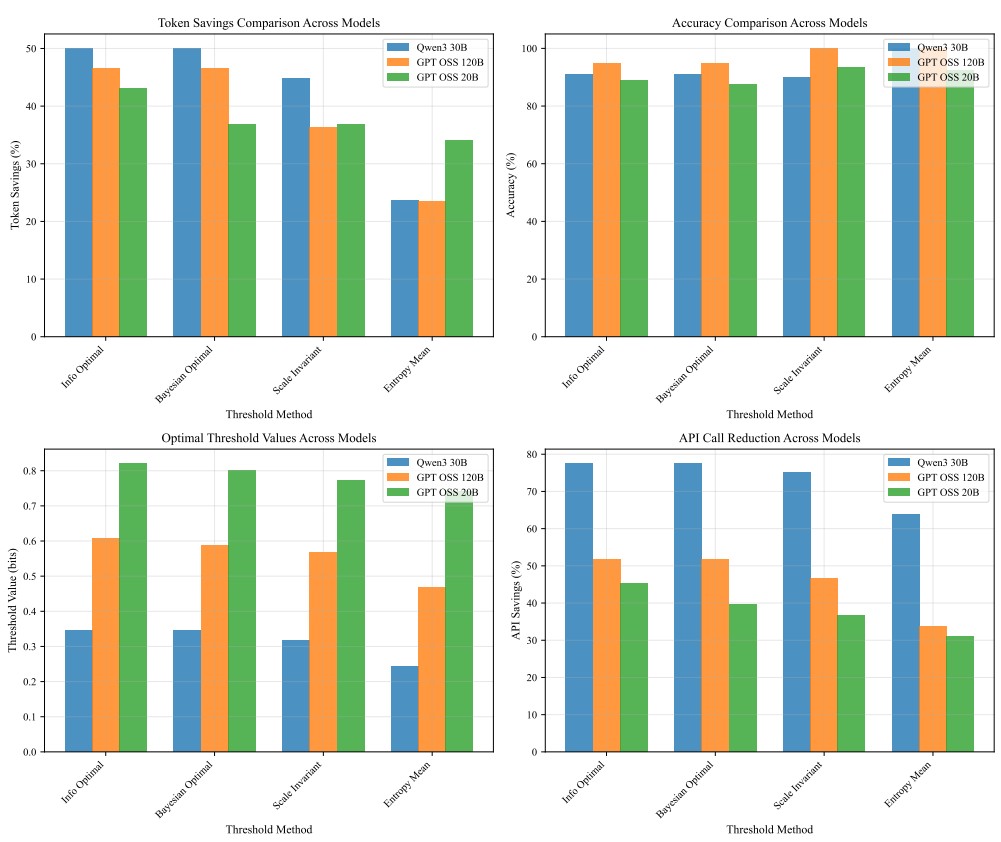

Figure 9: Comprehensive Pareto Analysis: (A) Accuracy-efficiency frontier showing ≥30% token savings with ≈0% accuracy loss region (highlighted in green). (B) Cross-model performance with Scale-Invariant Universal method. (C) Threshold method effectiveness comparison. (D) Cross-dataset robustness validation. All points include 95% confidence intervals for Δ-accuracy estimates.

