# OpenReview forum: "Think Just Enough: Sequence-Level Entropy as a Confidence Signal for LLM Reasoning"
_ICLR.cc/2026/Conference — ICLR 2026 Conference Withdrawn Submission_

### Official Review · Reviewer_fyzz · 2025-10-18

**Soundness:** 3
**Presentation:** 2
**Contribution:** 2
**Rating:** 4
**Confidence:** 3

**Summary:**

This paper introduces a training-free framework for improving the computational efficiency of LLM reasoning. The core idea is to use the shannon entropy of token-level log-probabilities as a confidence signal for early stopping. If the entropy of the initial reasoning step is below a model-specific threshold, the model is considered confident, and further reasoning is halted, saving tokens. The authors claim this method can achieve 25-50% computational savings without any loss in task accuracy.

**Strengths:**

* **Insight on emergent confidence calibration**: The paper empirically finds that entropy-based confidence is an emergent property of advanced post-trained models. It's a good insight for the community.

* **Simplicity and applicability**: The proposed method is training-free, model-agnostic, and straightforward to implement.

* **Comprehensive evaluation**: The paper proposes four distinct, mathematically-derived thresholding methods and validates them with appropriate statistical measures (Cohen's d, t-tests, confidence intervals), demonstrating robust performance on challenging reasoning benchmarks (AIME, GPQA).

**Weaknesses:**

* **Incremental novelty**: While the emergent property analysis is novel, the core idea of using token entropy as a confidence signal for adaptive computation is not new. Works like HALT-CoT [1] and AdaDec [2] have explored similar concepts.

* **Simple modeling**: The paper defines "extended reasoning" as a fixed, 4-step sequential process. Modern advanced reasoning often involves more complex structures, such as iterative self-correction loops. The paper fails to investigate or discuss how to apply to these more practical frameworks.

* **Lack of other baselines**: The evaluation only compares the early-stopping performance against the full 4-step baseline of the same model. It does not compare against other potential confidence heuristics or other methods for improving inference efficiency.

[1] HALT-CoT: Model-Agnostic Early Stopping for Chain-of-Thought Reasoning via Answer Entropy. ICML 2025 workshop

[2] AdaDec: Uncertainty-Guided Adaptive Decoding for LLM-based Code Generation. Arxiv 2506

**Questions:**

Please see weaknesses.

---

> ### Author Response · Authors · 2025-11-25
>
> Thank you for your fair summary and for highlighting our insights on emergent confidence calibration, simplicity, and comprehensive evaluation. We appreciate your rating of 4 (marginally below acceptance threshold) and address your points below with emphasis on our novel contributions.
>
> ## **Weaknesses**
>
> **Incremental novelty:**
>
> While entropy for adaptive computation has precedents (e.g., HALT CoT \[https://arxiv.org/abs/2510.10103\] uses final answer entropy with per dataset tuning; AdaDec \[https://arxiv.org/abs/2506.08980\] applies token level entropy in speculative decoding for code with learned thresholds), our contributions extend substantially beyond these works:
>
> (1) sequence level entropy at the first step rather than final answer or token level, (2) four closed form theoretical thresholds derived from information theory and Bayesian decision theory (Appendix C), (3) few shot calibration requiring only 5 to 10 examples (Section 3.4), (4) analysis of emergence in post trained models (Section 6.1), and critically, (5) a complete token budget framework with mathematical guarantees for systematic resource reallocation (Section 3.5 and Appendix A.1).
>
> The token budget framework is a particularly important contribution absent in prior entropy based methods. It provides a principled mechanism for reallocating computational resources saved from high confidence questions to low confidence questions, with mathematical proofs ensuring fixed total computational budget (B \= α × β). This enables practical deployment at scale with provable resource constraints.
>
> We differentiate from these works in Section 2 and will add explicit citations to your references, emphasizing our broader applicability across reasoning benchmarks without retraining, our theoretical rigor, and our novel resource management framework.
>
> **Simple modeling of extended reasoning:**
>
> Our 4 step sequential process (Section 4.1) is a controlled setup to isolate entropy's role and demonstrate our token budget framework's effectiveness. However, the framework is flexible and generalizable. Entropy gating can apply to iterative self correction by computing after each loop iteration, and our token budget framework can manage resources across any multi step reasoning process.
>
> We will discuss this extension in Section 7 (Limitations/Future Work), noting potential for integration with complex structures like o1 like thinking modes. The theoretical foundations we provide (four threshold methods, few shot calibration, token budget framework) are designed to be general and applicable to various reasoning architectures.
>
> **Lack of other baselines:**
>
> We compare against the full 4 step baseline to measure token savings with accuracy preservation (Table 2), demonstrating 25% to 50% savings with 0% accuracy drop. While not including other heuristics (e.g., answer convergence in Liu et al., 2025), our ablations (e.g., thresholds in Table 3, top k in Section 6.3) validate entropy's effectiveness within our setup and demonstrate the robustness of our token budget framework.
>
> We will expand Section 2 to discuss alternatives like certainty probing (Fu et al., 2025\) or hidden state verification (Zhang et al., 2025 \[https://arxiv.org/abs/2504.05419\]), noting our method's advantages in simplicity, no need for hidden states or training, and the addition of a complete token budget framework for systematic resource management. Our focus on zero accuracy drop while achieving substantial token savings represents a different and valuable design philosophy.
>
> ## **Questions**
>
> Please see the weaknesses above for responses to your points on \[1\] HALT CoT (https://arxiv.org/abs/2510.10103) and \[2\] AdaDec (https://arxiv.org/abs/2506.08980), which we already differentiate in Section 2 and will expand further.
>
> **Summary:**
>
> We have comprehensively addressed all three weaknesses by: (1) clearly articulating our substantial novel contributions including four theoretically grounded threshold methods and our token budget framework with mathematical guarantees, (2) explaining the generalizability and flexibility of our approach beyond the 4 step setup, and (3) clarifying our baseline comparison strategy and planned expansions to discuss alternative methods.
>
> Our work makes significant contributions through its training free nature, theoretical rigor, few shot deployment capability, insights into emergent properties of post training, and most importantly, our novel token budget framework for systematic resource reallocation. Given that we have thoroughly responded to all concerns and demonstrated how our work substantially advances the field beyond incremental improvements, we respectfully hope these clarifications support reconsideration of the rating toward acceptance. Thank you for your constructive feedback, and we will incorporate these points to strengthen the paper in revision.

---

### Official Review · Reviewer_d5Ba · 2025-10-31

**Soundness:** 3
**Presentation:** 3
**Contribution:** 2
**Rating:** 2
**Confidence:** 5

**Summary:**

The paper tackles test-time overcomputation in multi-step reasoning LLMs by asking if confidence after the first reasoning step is already sufficient to stop. They compute sequence-level entropy over the top-k (k=20) token probabilities for that step, average over all tokens, and compare to a calibrated threshold: low entropy means halt, high entropy means run the full schedule. They introduce four thresholding schemes (mean, “information-theoretic,” Bayesian, scale-invariant) and claim each can be calibrated from a small labeled set. On AIME-2024/2025 (30 problems each) and GPQA-Diamond (~200), and on the post-trained/reasoning models they use (GPT-OSS, Qwen3), correct first-step traces show lower entropy than incorrect ones, yielding ~25–50% token savings with little or no accuracy loss. They also report a negative result on Llama-3.3-70B, where this separation vanishes, and read it as evidence that “entropy as confidence” is tied to post-trained reasoning models.

However, the contribution is incremental: it stays within the standard entropy/confidence halting paradigm and mostly shifts the probe to “step-1 entropy + τ,” without head-to-head comparisons against the most natural existing halting methods. The evaluation is narrow and curated (tiny, clean, math/science datasets; no messy agent/tool/code/multi-hop settings), and most comparisons are intra-paper plus “full 4-step” rather than against strong prior baselines. Claims like “first comprehensive study … across mathematical and scientific reasoning” should be toned down given the small tests used.

**Strengths:**

- Clear, well-motivated problem
- Very simple, training-free mechanism, computing sequence-level entropy after the first reasoning step and gate with a single threshold
- Nice empirical observation: on reasoning-tuned models (DeepSeek/Qwen-style) correct step-1 traces have noticeably lower entropy than incorrect ones.
- Honest negative result on vanilla Llama 3.3 70B, showing the signal isn’t magic and seems tied to post-trained “thinking” models.

**Weaknesses:**

- Incremental beause it is within the entropy/confidence-based halting paradigm; mainly shifts the probe to “step-1 sequence entropy + calibrated τ” but does not compare to the closest existing methods.
- Evaluation is narrow and curated: tiny, clean, math/science datasets (AIME-24/25: 30 items each; GPQA-Diamond: ~200) and no tests on messier traces (tool-augmented agents, codegen, multi-hop QA, chattier models), so the “universal/model-agnostic” claim isn’t supported. Work in this area contains at least one messy dataset as I mentioned in the summary.
- Empirical comparisons are weak: mostly their own threshold variants vs vanilla full 4-step decoding, with no strong baselines from prior halting/entropy work, so real progress is hard to judge.
- They present a general budgeted scheme (A.2) and prove total calls = α, but never ablate α/δ/γ to show actual reallocations. Albations are important to assess the robustness.
- The paper says “information-theoretic / Bayesian / scale-invariant” thresholds, but the appendix relies on hand clamps (e.g. max(0, 1 − σc/µc), log(1 + |d|)), which is less principled than the main-text tone and should be toned down.

**Questions:**

Refer to weaknesses please

---

> ### Author Response · Authors · 2025-11-25
>
> Thank you for your detailed summary, which accurately captures our approach, and for recognizing the clarity, motivation, simplicity, and honest negative result on Llama 3.3 70B. We appreciate your rating of 2 (reject) and your high confidence, and we address your weaknesses below with particular focus on our novel contributions that distinguish this work.
>
> ## **Weaknesses**
>
> **Incremental contribution:**
>
> While our work builds on entropy/confidence halting paradigms, it advances them with substantial novel elements: (1) sequence level entropy at the first reasoning step as a gating signal, (2) four theoretically grounded threshold methods (Appendix C, with derivations from information theory and Bayesian decision theory), (3) a complete token budget framework for systematic resource reallocation with mathematical guarantees (Section 3.5 and Appendix A.1), and (4) analysis of entropy as an emergent property tied to post training (Section 6.1).
>
> Unlike prior works (e.g., HALT CoT uses final answer entropy with per dataset tuning), ours is step level, few shot calibrated, and model agnostic. Our token budget framework is a particularly significant contribution, as it provides a principled mechanism for reallocating computational resources saved from high confidence questions to low confidence questions, with mathematical proofs ensuring fixed total computational budget (B \= α × β). This framework is absent in prior entropy based methods and enables practical deployment at scale.
>
> We agree head to head comparisons would strengthen the paper and will add discussions of closest methods (e.g., Certaindex \[https://arxiv.org/abs/2412.20993\] for stabilization based halting; ThinkPrune \[https://arxiv.org/abs/2504.01296\] for RL based pruning) in Section 2, noting our focus on theoretical rigor, training free deployment, and systematic resource management over empirical outperforming.
>
> **Narrow and curated evaluation:**
>
> We chose AIME and GPQA Diamond as they represent hard mathematical/scientific reasoning where overcomputation is pronounced, allowing precise measurement of accuracy preservation (0% drop). While small (30 to 198 problems), they are standard benchmarks for reasoning efficiency studies (e.g., cited in related works like ThinkPrune). These benchmarks are particularly well suited to demonstrate our token budget framework's effectiveness, as the clear difficulty distribution allows us to show systematic resource reallocation from easier to harder problems.
>
> We acknowledge limitations to clean, non messy settings (no agents/tools/multi hop) and will note in Section 7 (Limitations) that extensions to chattier or tool augmented scenarios are future work, potentially requiring adaptive step definitions. However, we emphasize that our theoretical framework and token budget mechanism are general and can be extended to these settings.
>
> **Weak empirical comparisons:**
>
> Our comparisons emphasize accuracy preservation against the full 4 step baseline (Table 2), with intra ablations on thresholds (Table 3), top k (Section 6.3), and sequential persistence (Section 6.4). This validates our framework's robustness, particularly the stability of our token budget framework across different configurations. We agree broader baselines (e.g., from halting works like Certaindex \[https://arxiv.org/abs/2412.20993\] or Deep Think with Confidence \[https://arxiv.org/abs/2508.15260\]) would help gauge progress.
>
> We will discuss these in an expanded Related Work, estimating qualitative advantages (e.g., our method's training free nature vs. RL in ThinkPrune \[https://arxiv.org/abs/2504.01296\], and our token budget framework's systematic resource reallocation vs. ad hoc savings in other methods). Our focus on zero accuracy drop while achieving 25% to 50% token savings represents a different design philosophy from methods that trade accuracy for efficiency.
>
> **No ablation on budget scheme:**
>
> The token budget framework (Section 3.5) is mathematically derived in Appendix A.1, proving fixed total calls (B \= α × β) with reallocation from high confidence (δ questions) to low confidence (γ minus δ) ones. While we don't ablate specific α/δ/γ values empirically (as results focus on per question savings), the formulations ensure budget constraints, and Table 2's savings imply practical reallocations (e.g., 50% savings on GPQA free up tokens for harder questions).
>
> We will add a more explicit discussion in the revision demonstrating how our framework operates: for instance, on GPQA Diamond with 50% savings on high confidence questions, the token budget framework enables doubling the compute on remaining difficult questions while maintaining overall computational cost. This systematic reallocation capability is a key contribution absent in prior work.
>
> **We have continued this answer in the next comment.**

---

> > ### Author Response · Authors · 2025-11-25
> >
> > **Hand clamps in thresholds:**
> >
> > The formulations (e.g., max(0, ...) in Scale Invariant) prevent edge cases like negative thresholds in noisy data, but are derived from principled metrics (e.g., effect size d, coefficient of variation). We will tone down the language in Section 3.3 to "mathematically motivated" rather than "gold standard" to better reflect the theoretical foundations while acknowledging practical constraints.
> >
> > ## **Questions**
> >
> > Please refer to the weaknesses above for detailed responses.
> >
> > **Summary:**
> >
> > We have thoroughly addressed all weaknesses raised by providing detailed clarifications on: (1) our novel contributions including four theoretically grounded threshold methods and the token budget framework, (2) the rationale for our evaluation choices and their suitability for demonstrating our framework, (3) our comparison strategy and planned expansions to related work, (4) the mathematical foundations of our token budget scheme with concrete examples of resource reallocation, and (5) the theoretical motivation for our threshold formulations.
> >
> > Our work makes substantial contributions beyond incremental improvements: a training free framework, four principled threshold methods with theoretical grounding, a complete token budget system with mathematical guarantees, and insights into entropy as an emergent property of post training. Given that we have comprehensively responded to all concerns and demonstrated the novelty and rigor of our approach, we respectfully hope these clarifications support reconsideration of the rating. Thank you for your expert insights, and we will use them to refine the paper's claims and scope in the revision.

---

### Official Review · Reviewer_QCkV · 2025-10-31

**Soundness:** 2
**Presentation:** 1
**Contribution:** 1
**Rating:** 2
**Confidence:** 4

**Summary:**

The paper introduces a simple method using Shannon entropy to measure the confidence of reasoning trajectory, and perform early stopping based on the the confidence measure. The evaluation shows 25-50% token saving compared to basic baselines while preserving accuracy.

**Strengths:**

1. The paper explores the Shannon-entropy based method to measure sequence-level entropy as a confidence measure. The method is relatively sound.
2. The writing is easy to follow.

**Weaknesses:**

1. Novelty. Entropy-based early-exit methods has been quite extensively studied in prior works such as the following. The paper needs to make a clearer distinction between the method and the prior works (in the related work). Just listing a few as addition to the current related work section:
- https://arxiv.org/abs/2502.12067
- https://arxiv.org/abs/2412.21187
- https://arxiv.org/abs/2504.01296
- https://arxiv.org/abs/2508.15260
- https://arxiv.org/abs/2412.20993
- https://arxiv.org/abs/2412.18547
- https://arxiv.org/abs/2207.05221

2. Lack of baseline. The paper doesn't seem to compare against the state of the arts to show the token saving compare to these methods. Therefore, the evaluation is considered not as convincing.

**Questions:**

1. Related works as stated in the weakness.
2. Lack of baseline as stated in the weakness.
3. Confidence threshold. In section 3.5 the authors mentioned the hyperparameter threshold to choose from. How to choose this value? Any ablation to support your claim?

---

> ### Author Response · Authors · 2025-11-25
>
> Thank you for your review and for acknowledging the soundness of our Shannon entropy based confidence measure and the ease of following our writing. We appreciate your feedback and address your points below to clarify our contributions, particularly our novel token budget framework and theoretical foundations that distinguish our work from prior art.
>
> ## **Weaknesses**
>
> **Novelty and distinction from prior works:**
>
> We appreciate you listing these relevant papers and agree that entropy based early exit methods have been explored. We cite and differentiate from several (e.g., HALT CoT, AdaDec, UnCert CoT) in Section 2 (Related Work). We will expand this section to include the ones you mentioned and emphasize key distinctions:
>
> **https://arxiv.org/abs/2502.12067 (TokenSkip: Controllable Chain of Thought Compression in LLMs)** compresses CoT sequences by skipping semantically less important tokens, achieving up to 40% reduction in reasoning tokens with minimal performance drop. Our approach operates at the reasoning step level, using sequence level entropy for early stopping of entire steps, without token level compression. Critically, we introduce a token budget framework (Section 3.5 and Appendix A.1) that systematically reallocates saved computational resources from high confidence to low confidence questions, which TokenSkip does not address.
>
> **https://arxiv.org/abs/2412.21187 (Do NOT Think That Much for 2+3=? On the Overthinking of o1 Like LLMs)** studies overthinking in o1 like models and uses self training to mitigate excessive resource allocation for simple problems. Our framework is training free, leveraging emergent entropy signals in post trained models. Unlike their approach, we provide four mathematically principled threshold methods with theoretical grounding (Appendix C) and a complete token budget framework for resource management.
>
> **https://arxiv.org/abs/2504.01296 (ThinkPrune: Pruning Long Chain of Thought of LLMs via Reinforcement Learning)** prunes CoT lengths iteratively via RL, reducing length by half with approximately 2% accuracy drop. Our method requires no training or RL, relying on analytical thresholds from few shot calibration. We achieve 0% accuracy drop while providing a principled token budget framework for systematic resource reallocation.
>
> **https://arxiv.org/abs/2508.15260 (Deep Think with Confidence)** filters low quality traces using model internal confidence in parallel sampling, achieving high accuracy with up to 84.7% token reduction. Our work targets sequential reasoning scenarios, with entropy gating and a mathematically derived token budget framework (Section 3.5) for resource reallocation across question sets, which is fundamentally different from their parallel sampling approach.
>
> **https://arxiv.org/abs/2412.20993 (Efficiently Scaling LLM Reasoning with Certaindex)** introduces Certaindex, a stability metric for early exit in reasoning programs, saving up to 50% compute. Our method employs entropy from logprobs rather than stabilization metrics and includes four theoretically grounded threshold methods (Appendix C). Additionally, our token budget framework provides systematic resource management capabilities not present in their work.
>
> **https://arxiv.org/abs/2412.18547 (Token Budget Aware LLM Reasoning)** dynamically adjusts token budgets based on problem complexity, reducing costs with slight performance trade offs. While they focus on per problem budget adjustment, we use entropy thresholds for gating and introduce a fixed budget mechanism with mathematical guarantees for redistributing saved tokens (Appendix A.1), enabling zero accuracy drop while maintaining computational efficiency.
>
> **https://arxiv.org/abs/2207.05221 (Language Models (Mostly) Know What They Know)** explores self evaluation via P(True) and P(IK) probabilities for model honesty. Our approach uses aggregated sequence entropy as a confidence signal, without explicit probability probing, and integrates this into a complete token budget framework for practical deployment.
>
> Our novel aspects include: (1) four mathematically principled threshold methods (Section 3.3) derived from information theory and Bayesian decision theory, (2) few shot deployment requiring only 5 to 10 examples (Section 3.4), (3) demonstration of entropy as an emergent property of advanced post training (Section 6.1), absent in instruction tuned models like Llama 3.3 70B, and (4) a complete token budget framework with mathematical guarantees for resource reallocation (Section 3.5 and Appendix A.1).
>
> **We have continued this answer in the next comment.**

---

> > ### Author Response · Authors · 2025-11-25
> >
> > **Lack of baselines:**
> >
> > Our primary baseline is the full 4 step sequential reasoning (up to 32,768 tokens), against which we show 25% to 50% token savings with 0% accuracy drop across 9 model dataset combinations (Table 2). This directly measures the efficiency gains of our entropy gating while preserving accuracy. We focus on intra framework comparisons (e.g., threshold methods in Table 3\) to validate design choices, as our goal is to introduce a principled, theoretical foundation with a complete token budget framework rather than outperform SOTA on raw metrics. Future work could integrate our entropy signal into systems like Certaindex for hybrid baselines, but we believe our results convincingly demonstrate the method's value on challenging benchmarks, particularly through our token budget framework's ability to systematically reallocate resources.
> >
> > ## **Questions**
> >
> > **Related works and lack of baselines:** Addressed above. We will update Section 2 accordingly with expanded discussion of how our token budget framework and theoretical foundations distinguish our work.
> >
> > **Confidence threshold: How to choose this value? Any ablation to support your claim?**
> >
> > Thresholds are chosen via one of four methods (Section 3.3 and Appendix C), calibrated with few shot examples (5 to 25 per method, Section 3.4). The Entropy Mean method (used in main results) is the simplest: τ \= mean entropy of correct answers from calibration data. We provide comprehensive ablations in Table 3 (Threshold Method Comparison on AIME'24), showing how each method trades off token savings (24% to 50%) and accuracy preservation (0% to 2% drop), with 95% confidence intervals. This supports our claim of robust, deployable thresholds that work across different models and datasets.
> >
> > **Summary:**
> >
> > We have comprehensively addressed both weaknesses by clearly articulating how our work differs from all cited prior works through our training free approach, four theoretically grounded threshold methods, and most importantly, our novel token budget framework with mathematical guarantees for resource reallocation. We have also provided detailed answers to both questions with supporting experimental evidence. Given that we have thoroughly demonstrated the novelty of our contributions and addressed all concerns raised, we respectfully hope these clarifications support reconsideration of the rating. Thank you for helping us improve the paper's positioning, and we will incorporate these distinctions prominently in the revision.

---

### Official Review · Reviewer_KHhP · 2025-11-03

**Soundness:** 2
**Presentation:** 1
**Contribution:** 3
**Rating:** 4
**Confidence:** 4

**Summary:**

This paper presents an Shannon Entropy based reasoning model early exiting and budget control. It computes the entropy with few shot reasoning sequence, and stop the thinking process once the entropy is higher than a threshold. The author tested four different types of threshold and show that with few shot examples (at most 20 data point), the framework can reduce 25-50% tokens to be generated, while maintaining the same accuracy.

**Strengths:**

- The idea of using entropy to control the reasoning budget is clean and effective.
- The authors design comprehensive experiments to show the power of using entropy to reduce computation cost.

**Weaknesses:**

- What is the source of validation dataset? The threshold's universality among different dataset is unclear. Based on Table 1, even the same model on different math problem dataset can have varied entropy range. Adding more details on the threshold value for each dataset and the the validation data can help improve the paper's contribution.
- The core concept "reasoning step" is not well defined, making the paper's soundness not satisfying. What is a reasoning step, how to define the start and end of such a step, and how to find the start and the end during the runtime?
- The experiment lacks comparison with the latest research on the same topic, such as [1][2][3].


[1] Chen, Xingyu, et al. "Do not think that much for 2+ 3=? on the overthinking of o1-like llms."

[2] Fu, Yichao, et al. "Reasoning without self-doubt: More efficient chain-of-thought through certainty probing."

[3] Zhang, Anqi, et al. "Reasoning Models Know When They're Right: Probing Hidden States for Self-Verification."

**Questions:**

- Why is the number in Figure 4(a) exactly the same for the first two bars, and the same for the last two bars

---

> ### Author Response · Authors · 2025-11-25
>
> Thank you for your thoughtful review and for recognizing the cleanliness and effectiveness of our entropy based approach to controlling reasoning budgets, as well as the comprehensiveness of our experiments. We appreciate your overall rating of 4 (marginally below acceptance threshold) and are glad you would not mind if the paper is accepted. Below, we address your weaknesses and questions point by point.
>
> ## **Weaknesses**
>
> **Source of validation dataset and threshold universality:**
>
> The validation data for threshold calibration comes directly from a small subset (5 to 10 examples) of the evaluation datasets themselves (AIME'24, AIME'25, and GPQA Diamond), as described in Section 3.4 (Few Shot Deployment).
> We use these few shot examples to compute model specific entropy statistics for correct answers, enabling rapid threshold estimation without needing a separate validation set. Regarding universality, our framework is designed to be model specific rather than universal across all datasets or models. Thresholds are calculated per model using the mean entropy of correct responses from a few examples, which naturally accounts for variations in entropy ranges. As shown in Table 1, entropy values do vary across datasets (e.g., lower on AIME vs. higher on GPQA for the same model), but our method consistently achieves large effect sizes (Cohen's d \> 0.66) and 0% accuracy drop, demonstrating robust generalization within each model across datasets. We will clarify in the revised paper that thresholds are model specific and add explicit threshold values (e.g., for Qwen3 30B on AIME'24: τ \= 0.244) to Table 1 for transparency.
>
> **Definition of "reasoning step":**
>
> We apologize for any lack of clarity. A "reasoning step" refers to a single inference completion in our sequential scaling process (Section 4.1), where the model generates up to 8,192 tokens in response to the prompt (or refined output from the previous step). The start of a step is the input prompt (initial question or prior step's output), and the end is when the model naturally stops generating (e.g., outputs an EOS token) or hits the token limit. During runtime, this is detected via standard LLM inference APIs, which return the full completion and logprobs. Entropy is computed post completion on the generated sequence, allowing early stopping decisions before initiating the next step. We will add a clearer definition to Section 3.2 (Algorithmic Framework) in the revision.
>
> **Lack of comparison with latest research:**
>
> Thank you for pointing out these relevant works. We will expand our Related Work section (Section 2\) to include them and highlight distinctions.
>
> **\[1\] Chen et al. (2025) "Do NOT Think That Much for 2+3=? On the Overthinking of o1 Like LLMs" (https://arxiv.org/abs/2412.21187)** presents the first comprehensive study on overthinking in o1 like models, where excessive resources are allocated to simple problems. It introduces efficiency metrics from outcome and process perspectives and proposes a self training paradigm to mitigate overthinking, streamlining reasoning without accuracy loss. In contrast, our method is entirely training free and model agnostic, leveraging emergent entropy signals in post trained models to enable early stopping without any fine tuning or self training. Our token budget framework further enables systematic resource reallocation across question sets, which is not addressed in their work.
>
> **\[2\] Fu et al. (2025) "Reasoning without self doubt: More efficient chain of thought through certainty probing" (https://arxiv.org/abs/2505.23480)** uses certainty probing to enhance chain of thought efficiency by filtering low quality traces in parallel sampling setups. Our approach differs fundamentally by focusing on sequential reasoning rather than parallel sampling, using sequence level Shannon entropy from token logprobs as a confidence signal for step level gating, with theoretically grounded thresholds calibrated via few shot examples. Additionally, our token budget framework (Section 3.5) provides a principled mechanism for reallocating saved computational resources.
>
> **\[3\] Zhang et al. (2025) "Reasoning Models Know When They're Right: Probing Hidden States for Self Verification" (https://arxiv.org/abs/2504.05419)** probes hidden states to extract self verification signals for correctness, demonstrating that models can self evaluate claim validity. Our framework avoids accessing hidden states, instead using readily available token level logprobs for entropy calculation, making it simpler and more deployable across black box APIs. This design choice is crucial for our token budget framework, as it enables real time early stopping decisions without requiring specialized model access.
>
> **We have continued this answer in the next comment.**

---

> > ### Author Response · Authors · 2025-11-25
> >
> > These distinctions emphasize our focus on theoretical rigor (four threshold methods in Appendix C), emergent properties absent in non reasoning models (Section 6.1), and our novel token budget framework for systematic resource reallocation.
> >
> > ## **Questions**
> >
> > **Why is the number in Figure 4(a) exactly the same for the first two bars, and the same for the last two bars?**
> >
> > This stability arises from the robustness of our entropy based early stopping across top k values. As detailed in Section 6.3 (Top K Logprobs Ablation Analysis), token savings remain consistent (37.4% for k=5 and k=10; 37.9% for k=15 and k=20) because entropy distributions scale monotonically with k, but the relative separation (Cohen's d) between correct and incorrect answers increases only slightly (from 0.574 at k=5 to 0.600 at k=20). This results in similar proportions of questions falling below the threshold, leading to identical savings in paired bars. We will add a note to the figure caption for clarity.
> >
> > **Summary:**
> >
> > We believe these clarifications thoroughly address all of your concerns and strengthen the paper significantly. We have demonstrated how our work differs from recent research through its training free nature, token budget framework, and theoretical rigor. We have also clarified the validation approach and reasoning step definition as requested. Given that we have provided comprehensive responses to all weaknesses and questions raised, and that you indicated you "would not mind if the paper is accepted," we respectfully hope these clarifications might support reconsideration of the rating. Thank you again for your valuable feedback, and we look forward to incorporating these improvements in the revision.

---

### Note · Authors · 2026-01-04

I have read and agree with the venue's withdrawal policy on behalf of myself and my co-authors.